# Analysis of Tumor-Infiltrating T-Cell Transcriptomes Reveal a Unique Genetic Signature across Different Types of Cancer

**DOI:** 10.3390/ijms231911065

**Published:** 2022-09-21

**Authors:** Mabel Vidal, Marco Fraga, Faryd Llerena, Agustín Vera, Mauricio Hernández, Elard Koch, Felipe Reyes-López, Eva Vallejos-Vidal, Guillermo Cabrera-Vives, Estefanía Nova-Lamperti

**Affiliations:** 1Department of Computer Science, Universidad de Concepción, Concepción 4070409, Chile; 2Molecular and Translational Immunology Laboratory, Department of Clinical Biochemistry and Immunology, Faculty of Pharmacy, Universidad de Concepción, Concepción 4070386, Chile; 3Division of Biotechnology, MELISA Institute, Concepción 4133515, Chile; 4Centro de Biotecnologia Acuicola, Departamento de Biologia, Facultad de Quimica y Biologia, Universidad de Santiago de Chile, Santiago 9170022, Chile; 5Núcleo de Investigación Aplicada en Ciencias Veterinarias y Agronómicas, Facultad de Medicina Veterinaria y Agronomía, Universidad de Las Américas, Santiago 7500975, Chile; 6Data Science Unit, Universidad de Concepción, Concepción 4070409, Chile

**Keywords:** single-cell RNA-seq, cancer, T-cell, immunopathology, metabolic pathways, viral pathways

## Abstract

CD8+ and CD4+ T-cells play a key role in cellular immune responses against cancer by cytotoxic responses and effector lineages differentiation, respectively. These subsets have been found in different types of cancer; however, it is unclear whether tumor-infiltrating T-cell subsets exhibit similar transcriptome profiling across different types of cancer in comparison with healthy tissue-resident T-cells. Thus, we analyzed the single cell transcriptome of five tumor-infiltrating CD4-T, CD8-T and Treg cells obtained from different types of cancer to identify specific pathways for each subset in malignant environments. An in silico analysis was performed from single-cell RNA-sequencing data available in public repositories (Gene Expression Omnibus) including breast cancer, melanoma, colorectal cancer, lung cancer and head and neck cancer. After dimensionality reduction, clustering and selection of the different subpopulations from malignant and nonmalignant datasets, common genes across different types of cancer were identified and compared to nonmalignant genes for each T-cell subset to identify specific pathways. Exclusive pathways in CD4+ cells, CD8+ cells and Tregs, and common pathways for the tumor-infiltrating T-cell subsets were identified. Finally, the identified pathways were compared with RNAseq and proteomic data obtained from T-cell subsets cultured under malignant environments and we observed that cytokine signaling, especially Th2-type cytokine, was the top overrepresented pathway in Tregs from malignant samples.

## 1. Introduction

The main role of the immune system is to protect the body against infections and abnormal cell growth and one of the main players in the recognition of pathogenic antigens or neoplastic transformation are T-lymphocytes. This subset has been shown to be involved in the regulation of the immune response by operating both cellular and humoral immunity [1]. T-lymphocytes can be divided into two main subpopulations, CD8+ cytotoxic T-cells and CD4+ helper T-cells. CD8-T cells are characterized by inducing cell-mediated lysis during viral infection and malignancy [2,3], whereas CD4-T cells play an important role in the adaptive immune system by inducing a regulated effective response to pathogens, associated with a cytokine profile and the modulation of other subsets such as macrophages, B cells and NKs [4,5]. CD4-T cells can be subdivided into effector T-helper lineages such as Th1, Th2 and Th17 and regulatory T-cells (Tregs). Tregs are a subpopulation of CD4-T cells that maintain self-tolerance and modulate the immune system by controlling proinflammatory responses via different suppressive mechanisms [6].

In cancer, the cytotoxic responses from CD8-T cells and effector Th1 and Th17 cells have been considered protective in terms of tumor development [7]. By contrast, the presence of Th2 and Tregs has been associated with bad prognosis [8,9]. The balance between these effector and regulatory responses can be affected by cancer cells that promote phenotypic changes [10,11] and migration [12] of regulatory subsets that inhibit antitumor proinflammatory responses. Therefore, it is crucial to identify whether Tregs, CD4-T and CD8-T cells surrounding tumors exhibit exclusive specific genetic signature across different types of cancer. This information will lead us to identify potential mechanism by which the tumors command Tregs, CD4-T and CD8-T signaling pathways, which could be used to support the information required to propose future therapeutical tools in cancer [13,14].

During the last decades, the former T-cell subsets have been identified with the analytical method of flow cytometry by using fluorescent-labeled antibodies against proteins such as CD3, CD8, CD4, CD25, CD127 and FOXP3 [15]. This technique is widely used in clinical samples for the monitoring of these cells in cancer immunotherapies [16,17]. However, nowadays, novel genetic sequencing techniques have allowed the identification of T-cells based on their genetic signature, including the same markers classically used in flow cytometry. It has been shown that the identification of T-cells by single-cell RNA sequencing (scRNA-seq) can provide not only the identification of these cells, but also new biological pathways related with their function [18,19].

scRNA-seq allows to obtain a full genetic description of single cells in comparison with massive sequencing. However, scRNA-seq contains more noise than the analysis of massive sequencing [20], due to a greater amplification of the genetic material and a smaller number of samples. Despite that, methods aimed at reducing dimensionality and identifying subpopulations, as well as clustering methods from machine learning, have improved the analysis to get reliable single T-cell data [21].

In this study, we analyzed publicly available scRNA-seq data from CD4-T, CD8-T and Treg cells isolated from melanoma [22,23], breast [24,25], lung [26,27], colorectal [28] and head and neck [29] cancer. We identified common genes among tumor-infiltrating T-cell subsets. We compared these genes with the genetic profile of the same nonmalignant tissue resident subsets. For malignant-related T-cells subsets, results showed that 652 genes in CD4-T, 69 genes in CD8-T and 557 genes in Treg cells were common among the different tumors, but different from genes from nonmalignant samples. In terms of pathway analysis, specific gene discrimination between malignant and nonmalignant samples revealed unique immune response pathways in CD4-T, CD8-T and Treg cells associated with metabolism and immunoregulation. In addition, we validated the pathways for the tumor-infiltrating T-cells with RNA-seq and proteomic data with our in-house experiments under a malignant environment. Our data revealed that a Th2 signature commanded by the cytokine response in Tregs was one of the top pathways found in our analysis and it was a common response in melanoma, colorectal and oral cancer.

## 2. Results

Figure 1 illustrates the bioinformatics pipeline for the analysis and identification of exclusive T-cell subsets transcriptomic pathways in malignancy. Briefly, the scRNA-seq datasets from breast (GSE114727 and GSE75688), lung (GSE126030 and GSE99254), colorectal (GSE108989), melanoma (GSE72056 and GSE123139), and head and neck cancer (GSE103322) were obtained from the Gene Expression Omnibus (GEO) repository. Each dataset was analyzed separately as a digital expression matrix and transcripts per million values were used as gene expression levels (Figure 1A). Once the datasets were analyzed, we reduced dimensionality and clustered the different cell types in order to identify the T lymphocytes (Figure 1B). We then analyzed the transcriptome of CD4+, CD8+ and Tregs from malignant and nonmalignant samples (Figure 1C) and identified common genes across the different types of cancer for each T-cell subset (Figure 1D). Common genes from malignant and nonmalignant samples for each type of subpopulation were mapped to biological pathways using gene ontology (GO) terms [30,31], classified as biological process, molecular functions, cellular components and reactome pathways [32] (Figure 1E). The data were visualized using different networks (Figure 1F) and exclusive pathways for each subpopulation in both conditions were finally identified (Figure 1G).

### 2.1. Profiling of Tissue-Infiltrating T-Cells from Different Types of Cancer

In order to obtain the transcriptomic profile of tissue-infiltrating T cells, we analyzed each dataset individually to get the gene count for each cell type per condition (malignant, nonmalignant). We defined as a cutoff the upper median TPM values of gene expression in each scRNA-seq dataset and removed genes with low expression values in all the type of cells identified.

For datasets GSE126030, GSE99254 and GSE108989, the T-cells were already labeled in separate files in the Gene Expression Omnibus repository, therefore these files were used for a further analysis. Datasets GSE72056 and GSE103322 contained different types of cells embedded in a single matrix file, thus dimensionality reduction and clustering were required to identify T-cells. Briefly, we used PCA and variance analysis in order to obtain the value of the optimal number of components using the Scikit-learn implementation of PCA [33]. We identified eight components in melanoma (GSE72056) (Figure 2A) and nine components in head and neck cancer (GSE103322) (Figure 2B). In order to identify the T-cells population from these datasets, we performed a t-SNE dimensionality reduction and a clustering approach. Figure 2A,B show t-SNE plots identifying the different types of cells, including B cells, macrophages, endothelial, cancer-associated fibroblasts (CAFs), NK cells, mast cells and myocytes. Based on this analysis, only T-cells were selected by using the gene markers detailed in Appendix A.

For the identification of T-cell subsets in all datasets, we performed a comparison using a set of classical gene markers including *cd3g*, *cd8*, *cd4* and *foxp3*. Thus, these gene markers helped to filter and confirm the identification of CD4 helper T-cells (*cd3g* > 0; *cd8* = 0; *cd4* > 0; *foxp3* = 0), CD8 cytotoxic T-cells (*cd3g* > 0; *cd8* > 0; *cd4* = 0; *foxp3* > 0) and Tregs (*cd3g* > 0; *cd8* = 0; *cd4* > 0, *foxp3* > 0) from the original datasets. In Table 1, we reported a summary of the counting of genes by dataset after removing genes with low expression values, identifying the type of tissue with its respective Gene Expression Omnibus ID (GEO), and the counting of malignant- and nonmalignant-related genes for CD4-T, CD8-T and Treg cells. We identified a total of 67,917 genes in malignant CD4, 63,038 genes in malignant CD8 and 56,827 genes in malignant Treg from five tissues (breast, lung, colorectal, head and neck and melanoma). On the other hand, we identified 24,436 genes in nonmalignant CD4, 28,341 genes in nonmalignant CD8 and 20,877 genes in nonmalignant Treg for two tissues (lung and colorectal).

Once the different T-cell subsets were identified, we performed a second analysis to determine those common genes for each condition (malignant, nonmalignant) across all different type of samples (Figure 2C,D). Then, we compared the common genes between the malignant and nonmalignant condition for each T-cell subset. The Venn diagram in Figure 2E–G and Appendix A represent the total number of genes from scRNA-seq data for the CD4-T, CD8-T and Tregs subsets, respectively. We observed 652 (11.74%) exclusive genes in malignant-derived CD4-T cells, 69 (1.10%) exclusive genes in malignant-derived CD8-T cells and 557 (12%) exclusive genes in the malignant-derived Tregs analyzed. This overall gene identification revealed exclusive and common genes per T-cell subset and condition, allowing us to use this information for a further pathway analysis.

### 2.2. GO Annotations and Biological Pathways in T-Cells from Malignant and Nonmalignant Cancer

GO annotations and biological pathways were analyzed by comparing the data obtained from malignant and nonmalignant samples per subset. The Gene Ontology Consortium [30,31] was used to establish the GO enrichment and the reactome pathway database [32] to analyze pathways and generalize the concept of the reactions that matched our datasets, including biological processes with a focus on signaling, metabolism, transcriptional regulation, apoptosis and disease. We identified 7490 GO annotations for biological process, 1404 GO annotations for molecular functions, 2035 for GO annotations of cellular component ontology and 12,033 reactome pathways (Table 2). These annotations were divided into malignant and nonmalignant origin in CD4 (Appendix A), CD8 (Appendix A) and Treg cells (Appendix A). For GO terms 38, 72 and 100 biological functions were associated exclusively to malignant CD4-T, CD8-T and Treg cells, respectively. In general, by observing the Appendix A, from malignant samples we observed that some of the main associations found in the GO terms among CD4-T, CD8-T and Treg cells corresponding to the immune response and defense, some of them being transcendental for cancer immunotherapies [34,35].

In order to determine the interaction between biological pathways associated with malignant or nonmalignant samples and visualize them as an interaction network, the biological process terms were analyzed using Cytoscape software [36] and the ClueGO plugin [37] to visualize the nonredundant biological terms for large clusters of genes as a hierarchical biological network. Using this method, we observed 17 exclusive GO terms in malignant CD4, 17 exclusive GO terms in nonmalignant CD4, 12 exclusive GO terms in malignant CD8, 86 exclusive GO terms in nonmalignant CD8, 54 exclusive GO terms in malignant Treg and 25 exclusive GO terms in nonmalignant Treg cells. Figure 3 displays a visualization of the network for malignant and nonmalignant genes for CD4-T, CD8-T and Treg cells, including clustered pathways for each condition (malignant and nonmalignant) and labeling according to the most significant term per group (Supplemental Networks available at https://github.com/tumourTcells/Cytoscape-file (accessed 10 March 2021)). For the specificity of each pathway, we observed differences between malignant and nonmalignant samples. These data are available in the child nodes that, together with the *p*-values information from ClueGO, indicated different functional categories inside the networks. Of note, common functions between malignant and nonmalignant T-cell subsets were observed in this analysis despite the previous gene selection. In summary, our data revealed 93 clusters for malignant CD4, 92 clusters for nonmalignant CD4, 43 clusters for malignant CD8, 123 clusters for nonmalignant CD8, 83 clusters for malignant Treg and 81 clusters for nonmalignant Treg samples (Figure 3 and Appendix A). One cluster that was differentially observed in Treg cells was the regulation of macromolecule metabolic process, which was positive in malignant samples, but negative in nonmalignant Treg samples. Besides clusters, this analysis showed a different specialization among the pathways for each condition.

### 2.3. Exclusive Biological Pathways in T-Cells from Different Types of Cancer

After defining the pathways in cells from malignant and nonmalignant samples we identified exclusive pathways for T-lymphocytes between both conditions (Figure 4A and Appendix A). The 31 recurrent annotations found in the three T-cell subsets from malignant samples were associated with different types of cancer and other diseases of the immune system, the malignant genes being associated with the types of cancer studied and viral diseases such as HIV Infection. However, we also found that pathways associated with localization were common among the three subpopulations from malignant samples, such as a positive regulation of protein localization to Cajal bodies and SRP-dependent cotranslational protein targeting the membrane. Other important functions found were viral transcription and viral gene expression pathways, which play an important role in viral transcription and translation [38,39]. From nonmalignant samples, we observed 23 recurrent annotations within the T-cells highlighting the regulation of translation in response to stress, nucleotide-excision repair, DNA damage recognition and viral budding, which were the most important and common terms among T-cell subsets.

When molecular function annotations were analyzed, we observed that peroxiredoxin activity, threonine-type peptidase activity and NADH dehydrogenase activity were recurrent and relevant annotations for CD4, CD8 and Treg cells from malignant origin. For T-cells from nonmalignant origin, we found that among the three subpopulations’ structural constituent of ribosome, thyroid hormone receptor binding and snoRNA binding were common annotations among them. In the case of cellular components annotations, we observed that signal recognition particle, endoplasmic-reticulum-targeting and proteasome core complex and beta-subunit complex were the three most important annotations in common among malignant CD4, CD8 and Treg cells. For nonmalignant cells, we observed that methylosome, U2-type catalytic step 2 spliceosome and eukaryotic translation initiation factor 3 complex (eIF3m) were the most important annotations in common among CD4-T, CD8-T and Treg cells (Appendix A).

Exclusive functions of the different T-cell subsets between malignant and nonmalignant samples were finally analyzed. A total of 38 exclusive biological process terms belonged to malignant CD4 and 33 to nonmalignant CD4 cells, 72 exclusive biological process terms belonged to malignant CD8 and 80 to nonmalignant CD8 cells and 100 exclusive biological terms belonged to malignant Treg and 38 to nonmalignant Treg cells (Figure 4A). Figure 4B,C show the gene ontology terms associated with exclusive genes in two-dimensional scatter plots where the GO ID represents a semantic correlation from the gene ontology database and the *y*-axis represents the negative log *p*-value associated with genes and functions (Figure 4B,C).

The main pathways associated exclusively with malignant CD4 cells were the somatic diversification of immune receptors via germline recombination within a single locus, protein targeting the vacuole, lymphocyte activation involved in immune response, positive regulation of DNA-binding transcription factor activity, T-cell differentiation involved in the immune response, negative regulation of extrinsic apoptotic signaling pathway via death domain receptor and the positive regulation of leukocyte activation, among others. On the other hand, the main exclusive pathways for nonmalignant CD4 cells were the negative regulation of response to wounding, aminoglycan biosynthetic process, chemical synaptic transmission, cell communication involved in cardiac conduction, neuromuscular synaptic transmission, interleukin-15-mediated signaling pathway and the negative regulation of lymphocyte apoptotic process, among others.

Malignant CD8 was characterized mainly by metabolic process such as purine nucleoside triphosphate biosynthetic process, ATP biosynthetic process, response to epidermal growth factor, gluconeogenesis, response to gamma radiation, the negative regulation of oxidative stress and the positive regulation of signal transduction by p53 class mediator. Nonmalignant CD8 pathways were characterized by cell morphogenesis involved in differentiation, mitochondrial RNA metabolic process, DNA modification, phospholipid metabolic process, multicellular organism development, cell recognition and others.

In the case of malignant Tregs, we observed a detailed specificity in the pathways such as the positive regulation of cell differentiation, circadian regulation of gene expression, cellular response to glucocorticoid stimulus, response to drug, negative regulation of mRNA catabolic process, negative regulation of cell population proliferation, positive regulation of cell cycle G1/S phase transition, negative regulation of NIK/NK-kappaB signaling and tricarboxylic acid metabolic process, whereas for nonmalignant Tregs, the main pathways were the negative regulation of cytokine production, negative regulation of immune effector process, positive regulation of pathway-restricted SMAD protein phosphorylation, positive regulation in response to stress, positive regulation of T-cell proliferation and the negative regulation of receptor signaling pathway via JAK-STAT, among others. A complete detail of pathway description is available in Appendix A.

Across the T-cells subpopulations, we also observed 59 exclusive positive and negative regulations of different biological process. In Figure 5, we highlighted negative biological functions with a left orientation and positive biological functions with a positive orientation. Analyzing the different mechanisms of regulation from the cell helped us to understand how the regulation was working simultaneously in several pathways either positively or negatively to regulate the exclusive process of each T-cell subpopulation.

### 2.4. Reactome Pathways in T-Cells from Different Types of Cancer

Reactome pathways were also analyzed among T-cell subsets through the Reactome Pathway Database [32]. We observed a total of 5771 annotations for malignant data and 6262 annotations for nonmalignant data. In the three T-cell subsets derived from malignant samples, the main pathways found were associated with cellular response to stress, translation and ER–phagosome pathway. The nonmalignant derived T-cell subsets were also associated with RNA process and class I MHC-mediated antigen processing and presentation (Appendix A). In malignant CD4 cells, the most overrepresented pathways were SRP-dependent cotranslational protein targeting the membrane, the GTP hydrolysis and joining of the 60S ribosomal subunit and L13a-mediated translational silencing of ceruloplasmin expression, which are part of the metabolism of proteins in the database. In the case of nonmalignant CD4 cells, the main three pathways were translation, the metabolism of RNA and the processing of capped intron-containing pre-mRNA, thus regulating more processes of the metabolism of RNA. For malignant CD8, the most overrepresented pathways were peptide chain elongation, the formation of a pool of free 40S subunits and eukaryotic translation elongation, which are also part of the metabolism of proteins. In nonmalignant CD8, the same pathways as in nonmalignant CD4 cells were the most overrepresented. In malignant Treg, the formation of a pool of free 40S subunits, nonsense mediated decay (NMD) independent of the exon junction complex (EJC) and peptide chain elongation were the most important pathways, also related to the metabolism of protein and RNA. In nonmalignant Treg, we identified translation, the metabolism of RNA and regulation of expression of SLITs and ROBOs, observing in this subpopulation more pathways from the developmental biology location.

Furthermore, for a more exhaustive analysis and subsequent discussion we focused on the annotations under the immune system pathway, observing that TRAF6-mediated IRF7 activation in TLR7/8 or TLR9 signaling was present only in malignant CD4 cells. In nonmalignant CD4 cells, we also observed unique pathways that were the TLR3-mediated TICAM1-dependent programmed cell death, TICAM1-dependent activation of IRF3/IRF7, TICAM1, TRAF6-dependent induction of TAK1 complex and TICAM1, RIP-mediated IKK complex recruitment.

In summary, our data revealed that there were several exclusive pathways in T-cell subsets that were common among different types of cancer; however, in order to validate this information and point out targeted pathways, proteomics and RNA-seq data from our experiments in patients with oral cancer were used to complement our in silico analysis.

### 2.5. Validation of Biological Pathways Using Proteomics and RNA-seq Data

The classic phenotype of T-cells was used to identify the changes in the repertoire of Th subsets in different tumors [40]. Previous data from our lab demonstrated an increment in Th2-like Tregs- and Teff-infiltrated subsets in melanoma [6], colorectal cancer [6] and oral cancer [41] in comparison with infiltrated subsets from nonmalignant tissues (Figure 6A). In addition, we identified that Vitamin D signaling promoted these Th subset disbalance in oral cancer [41]. We then compared our RNAseq data obtained from T-cells subsets cultured with malignant and nonmalignant environments from oral cancer and proteomic data from CD4-T cells cultured with Vitamin D with the data obtained from the in silico analysis.

In total, 481 genes were obtained from the RNA-seq experiments. For CD4-T cells, 218 common pathways were identified between scRNA-seq and RNA-seq data, whereas for Tregs, 194 common pathways were identified. No CD8 cells were analyzed for RNA-seq data. For proteomics, only CD4-T cells were analyzed. Normalized intensities and normal distribution (Appendix A) were obtained from each condition. A principal component analysis showed a great similarity among replicates and a great difference at the level of differential protein expression among conditions (Appendix A). In total, 2558 proteins with statistically significant differential expression were obtained with an adjusted *p*-value of less than 0.05 (Appendix A) and a total of 4410 quantifiable proteins were represented in the volcano plot (Appendix A). After analysis, 1692 proteins were found in the gene list obtained from the sc-RNAseq data, resulting in 561 common pathways.

Signaling by interleukins, nucleotide-binding domain, leucine-rich repeat-containing receptor (NLR), TRAF6-mediated NF-kappaB activation and toll-Like receptor cascades were the top overrepresented common pathways in CD4-T cells (Figure 6B). For Tregs, the data revealed that signaling by interleukins (IL-4, IL-13 and IL-1-0), transcriptional regulation by TP53, interferon alpha/beta signaling, and nerve growth factor (NGF)-stimulated transcription and NTRK were the top overrepresented common pathways (Figure 6C). Finally, proteomic data revealed common pathways such as cytokine signaling in immune system, interferon gamma signaling, downstream TCR signaling, MHC class II antigen presentation and PD-1 signaling (Figure 6D). Overall, these data support the observation of a preference of regulatory Th2-like cells in cancer, as previously described by us and others, and other interesting pathways such as the recognition of bacteria and virus by NLR and TLR signaling in CD4 and the NGF receptor tyrosine-kinase TrkA signaling in Tregs.

We validated and observed common pathways for CD4-T cells and Treg with our experiments, maintaining all the discoveries from the in silico analysis across different types of cancers. A complete list of the pathways is available in Appendix A.

## 3. Discussion

Over the past years, datasets obtained from scRNA-seq have revealed valuable information about the repertoire of cells contributing or controlling the development of several types of tumors [22]. In this study, we analyzed published scRNA-seq datasets obtained from T-cell subsets from malignant and nonmalignant origin, in order to identify common pathways among these cells across different types of tumors. Our data revealed that regardless of the type of cancer, we observed common functions associated with metabolic process (Appendix A), translation and immune-related pathways for each T-cell subset. Publicly available data were analyzed through an exhaustive in silico analysis where samples were characterized and relabeled in order to generate a list of pathways and gene ontology (GO) categories, which were further validated by using data from different sources. Finding subgroups of cells and analyzing in tumor tissue versus normal tissue would lead us to understand common pathways of T-cells in different cancer. In addition, regardless of the tumor origin, similar pathways were identified; therefore, potential immunomodulatory therapies associated with these pathways could be applied to different types of tumors.

One of the major limitations in biological analysis is often the high dimensionality of the data [42]. Common machine learning methods have been applied to scRNA-seq data in order to reduce their dimensionality and identify subpopulations. Such methods include principal component analysis (PCA) [43], which aims at reducing the data dimensionality by calculating a transformation of the data into a set of linearly uncorrelated values called principal components. PCA is a simple and very useful tool for examining the heterogeneity in scRNA-seq data [44,45,46]. Recently, t-distributed stochastic neighbor embedding (t-SNE), has also been applied to dimensionality reduction in scRNA-seq data [22,47,48]. t-SNE is a stochastic method for dimensionality reduction originally aimed at visualizing high-dimensional data. It is a nonlinear dimensionality reduction technique that finds a lower dimensional space in which similar objects are close and dissimilar objects are distant with high probability [49]. Clustering techniques can then be applied to the reduced dimensionality space in order to find groups of similar cells [21]. In our study, it was essential to define a strategy to thoroughly label the samples, and thus ensure a more precise and reliable result by using machine learning techniques. Using the classification of cells in terms of their immunological background, we showed that a high similarity existed for the relevant genes. These genes were mapped to the same biological functions, mainly cancer-development functions, and finally, GO terms and reactome annotations gave us a clear idea of the pathways highlighted to define possible targets to identify key cellular pathways from the immune system in cancer.

A visualization of the overrepresenting GO annotations in T-cell subpopulations isolated from both malignant and nonmalignant tissues, allowed us to identify common genes between healthy and tumor-infiltrating T-cells, and also common genes across different types of cancer for each T-cell subset. We observed the highest percentage of common GO annotations between malignant and nonmalignant conditions in CD4-T cells. The most overrepresented terms found only in malignant CD4-T cells was the detection of an abiotic stimulus and the detection of an external stimulus that participates in the perception of the stimulus. Then, it is received by a cell and converted into a molecular signal [31,50,51]. The positive regulation of cysteine-type endopeptidase activity was also an overrepresented term in malignant CD4-T. This function is involved in apoptotic processes and inflammasomes, being also responsible for the activation of the inflammatory response [52]. The data also highlighted the positive regulation of the antigen-receptor-mediated signaling pathway as an important term in malignant helper cells. In fact, it has been associated with key immunological function in ovarian cancer among the four stages of this type of cancer, because it is one of the initial triggers of the immune response and can activate the T-cell response [53].

We also observed some overrepresented functions in nonmalignant CD4-T cells that indirectly could be important for cancer, as the absence of these may also contribute to tumor development. Here, the regulation of the oxidative-stress-induced intrinsic apoptotic signaling pathway, autophagosome maturation and the regulation of the vascular permeability process were overrepresented only in nonmalignant CD4-T cells. The first function plays an essential regulatory role in promoting cell survival under stress conditions contributing to cancer therapy [54]. For autophagosome maturation, this pathway is crucial in the delivery of cytoplasmic components. Therefore, the role in cancer for damaged proteins and organelles autophagy allows a prolonged survival of tumor cells, providing a protective function limiting tumor necrosis and inflammation [55,56]. In the case of vascular permeability, this pathway is related with blood distribution to all tissues and maintaining the homeostasis, lipid transport and immune surveillance. In the particular case of cancer, this permeability can facilitate metastatic spread [57]. Furthermore, vascular permeability is crucial in physiological and pathological angiogenesis, due to normal or healthy blood vessel growth occurring during tissue repair, and it has been reported as a cause of mortality in cancer, among other causes [58].

Overrepresented annotations in CD8-T cells from malignant samples were characterized by a development of biosynthetic processes, showing pathways such as purine nucleoside triphosphate biosynthetic process, ribonucleoside triphosphate biosynthetic process, purine ribonucleoside triphosphate biosynthetic process and ATP biosynthetic process. All these functions in cancer are associated with metabolic requirements for the cell growth and proliferation of cancer cells by producing de novo nucleotide synthesis, maintaining normal triphosphate biosynthetic process, as this process is critical for the replication and repair of the DNA [59,60]. Another pathway that characterized unique CD8-T responses in malignant conditions was the response to epidermal growth factor, which has been associated with the regulation of cell proliferation, differentiation and migration through the epidermal growth factor receptor (EGFR) function, which plays an important role in tumorigenesis in various types of epithelial cancers. Nowadays, novel therapies that target the EGFR agents have improved patient’s therapies with colorectal, lung, head and neck and pancreatic cancers; however, there are some cases using monoclonal antibodies where an activation of signaling pathways downstream of the EGFR could produce resistance to the treatment [61,62]. Gluconeogenesis was also observed only in malignant CD8-T cells, but the role of this metabolic process in CD8-T cells in cancer is unclear. It is known that it generates free glucose from precursors and is associated with cancer cell plasticity and tumor cell growth. However, it has also been shown that this pathway is inhibited in some types of cancers as it may engage in truncated gluconeogenesis function in fasting conditions [63,64,65]. On the other hand, overrepresented annotations in nonmalignant CD8-T were characterized by membrane invagination, multicellular organism development and inner ear morphogenesis and anatomical structure development that are associated with developmental processes. Those terms in general are part of the membrane organization and developmental processes. Others highlighted pathways observed were glycerolipid biosynthetic process, glycerophospholipid biosynthetic process and lipid biosynthetic process. Glycerolipid processes have been proposed within a new therapy in cancer, neuroscience and metabolic diseases by targeting them with small molecule inhibitors [66,67].

In Tregs, the pathways with the highest *p*-value from malignant samples were the positive regulation of cell differentiation and cellular response to different compounds, such as nitrogen, oxygen-containing, glucocorticoids stimulus and organonitrogen, among others. Those compounds in cancer studies have been proved, when altered, to support cancer and immune cells responses [68]. In the case of glucocorticoids, those corticosteroids act primarily on carbohydrate and protein metabolism having anti-inflammatory and immunosuppressive effects [69]. Other pathways associated only to malignant Tregs were the circadian regulation of gene expression and circadian rhythm, both processes modulating the frequency of gene expression pattern with a regularity of approximately 24 h. In cancer studies, those pathways participate in cyclic physiological processes. In addition, cancer has been linked with the disruption of circadian rhythms [70,71]. The regulation of NIK/NF-kappaB signaling, negative regulation of I-kappaB kinase/NF-kappaB signaling and negative regulation of NIK/NF-kappaB signaling were also associated only to malignant Tregs, affecting a complex network among extracellular stimuli to cell survival, developing an essential role in inflammation, innate immunity and cancer initiation and progression [72,73,74]. Nonmalignant Tregs present annotations such as retrograde vesicle-mediated transport, Golgi to endoplasmic reticulum, the maintenance of protein location and positive regulation of transport, which belong to localization and transport pathways. Furthermore, we observed pathways associated with cytokines such as the positive regulation of response to cytokine stimulus and the positive regulation of lymphocyte proliferation, which play an important role in the modulation of immune and inflammatory responses, due to cytokines’ key role in clinical cancer research [75,76,77,78].

Finally, the terms obtained from the Reactome analysis revealed pathways that play an important role in the immune system, such as cytokines that regulate and mediate immunity, inflammation and hematopoiesis, promoting intercellular communication among immune cells [79]. In addition, these proteins bind to their cell surface receptor and act in an autocrine and/or paracrine fashion, inducing tissue growth and repair [80,81]. Moreover, adaptive immunity is involved in roles such as the recognition of particular pathogens or antigens’ prior presentation by antigen-presenting cells from peripheral tissues [82]. TNFR2 noncanonical NF-kB pathway was one of the most significant pathways among all the datasets analyzed, even though it was found in both conditions (malignant and nonmalignant). It has been shown that TNFR2 in normal tissues exhibits a basal expression [83], whereas it can be induced to promote cell survival pathways, such as cell proliferation, by activating transcription factor NF-kB via the alternative noncanonical route [84]. This suggests that the regulation of TNFR2 may be relevant in tumor-infiltrating lymphocytes.

## 4. Materials and Methods

### 4.1. Data Collection and Preprocessing

Data collection from previous scRNA-seq expression profiles from malignant and nonmalignant cells were included in the analysis. In some cases, nonmalignant samples were obtained from the adjacent normal tissues. We selected scRNA-seq data from isolated cells from breast (GSE114727 and GSE75688), lung (GSE126030 and GSE99254), colorectal (GSE108989), melanoma (GSE72056 and GSE123139), and head and neck cancer (GSE103322). The datasets were obtained from the Gene Expression Omnibus (GEO) repository and all of them were sequenced on Illumina HiSeq2500/HiSeq4000 or Illumina NextSeq 500 (Homo sapiens) with a similar experimental design. We verified the quality of each sequencing library with FastQC [85], a software package that estimates the number of uncallable and low-quality bases. Mapping to the human reference genome (hg38) was done using STAR [86], a high-performance community-standard aligner.

Each dataset was analyzed separately as a digital expression matrix. We used transcripts per million (TPM) values as gene expression levels for all the analysis, calculated as:(1)$$\frac{10^{6}\cdot C_{ij}/\mathrm{length}\;\mathrm{of}\;\mathrm{gene}\;i}{\sum_{i}C_{ij}/\mathrm{length}\;\mathrm{of}\;\mathrm{gene}\;i},$$
where $C_{ij}$ is the count value of gene *i* in cell *j*. We removed genes with low expression values, considering as cutoff the upper median TPM values [87].

### 4.2. T-Cell Identification

Datasets GSE126030, GSE99254 and GSE108989 contained only T-cells. Datasets GSE114727, GSE75688 and GSE123139 contained different type of cells, although each of them was detailed in separate files available in the Gene Expression Omnibus (GEO) repository, indicating explicitly which of them were T-cells.

To identify the different cells for the datasets GSE72056 and GSE103322, we started by using PCA and variance analysis in order to obtain the value of the optimal number of components using the Scikit-learn implementation of PCA [33]. We identified eight components in GSE72056 and nine components in GSE103322.

We used agglomerative clustering to define subsets of cells and for assigning their labels (T, B, macrophages, endothelial, cancer-associated fibroblasts (CAFs) NK, mast and myocytes cells). We used t-SNE [88] for dimensionality reduction in order to visualize the cells in a two-dimensional scatter plot. Following [22,29], we used six cluster for GSE72056 and eight for GSE103322 (https://github.com/tumourTcells/codes, accessed on 20 March 2021) in order to cluster the same number of cells.

To identify the T-cell cluster in GSE72056 and GSE103322, we removed all the cells that had no expression for cell markers described in Section 2.1 (Appendix A) and those were the ones we used in the analysis.

### 4.3. Analysis of T-Cells Subpopulations

In order to identify our target genes, we classified the subpopulation of T-cells according to the following gene selection criteria: (1) for CD4-T cells, *cd3g* > 0, *cd8* = 0, *cd4* > 0 and *foxp3* = 0; (2) for CD8-T cells, *cd3g* > 0, *cd8* > 0, *cd4* = 0, *foxp3* > 0; and (3) for Tregs, *cd3g* > 0, *cd8* = 0, *cd4* > 0 and *foxp3* > 0.

To identify the T-cell subpopulations included in our analysis, we generate a multiple list comparator with the name of the genes in each condition, to finally extract which genes were common across malignant samples, nonmalignant samples and different between malignant and nonmalignant samples.

### 4.4. Proteomic Experiments

A total of $2\times 10^{5}$ CD4-T cells from healthy donors were activated with anti-CD3/CD28 beads (1:5 ratio) (Life Technologies) in XVIVO-15 media for 5 days a 37 °C in the presence or absence of 1,25(OH)VitD (10 nM in ethanol) or carrier (ethanol) as previously described [41]. After 5 days, the cells were washed and stored with protease inhibitor before the proteomics analysis.

Samples were analyzed with a nanoELUTE (Bruker Daltonics, Bremen, Germany) ultrahigh-pressure nanoflow chromatography system coupled online to a hybrid trapped ion mobility spectrometry–quadrupole-time-of-flight mass spectrometer (timsTOF Pro, Bruker Daltonics, Bremen, Germany) with a modified nanoelectrospray ion source (CaptiveSpray, Bruker Daltonics). Liquid chromatography was performed (50 °C, 400 nL/min constant flow on a reversed-phase column Aurora Series CSI (25 cm × 75 μm i.d. C18 1.6 μm) (ionopticks Australia)). Mobile phases A and B were watered with 0.1% formic acid (*v*/*v*) and 99.9/0.1% ACN/formic acid (*v*/*v*), respectively. Tandem mass spectra were extracted by Tims Control version 2.0. Charge state deconvolution and deisotoping were not performed. All MS/MS samples were analyzed using MSFragger 3.2 [89,90] set up to search the UniProt_SwissProt database (unknown version, 21,040 entries) assuming the digestion enzyme trypsin. MSFragger was searched with a fragment ion mass tolerance of 0.050 Da and a parent ion tolerance of 50 PPM. The carbamidomethyl of cysteine was specified in MSFragger as a fixed modification, the quantification used IonQuant for Label Free [91], and the peptide validation used Peptide Prophet [92].

The quantitative (LFQ) and statistical analysis used perseus software version 1.615 (Max Planck Institute, Martinsried, Germany) [93].

In order to clarify the source and origins of malignant and nonmalignant T-cell subsets please refer to Halim et al. [6] and Fraga et al. [41].

### 4.5. Analysis of RNA-seq

The same preprocessing as that of the scRNA-seq samples was used. We verified the quality of each sequencing library with FastQC [85], and the reads were aligned to the human reference genome (hg38) using STAR [86]. We used transcripts per million (TPM) values as gene expression levels. The samples are available in the GEO repository with the accession identifier GSE171638. In order to clarify the source and origins of malignant and nonmalignant T-cell subsets, please refer to Halim et al. [6] and Fraga et al. [41].

### 4.6. Pathways and GO Categories Analysis

A pathway enrichment analysis was performed using the Gene Ontology Consortium database (data-version from 2 May 2020) [30,31]. This database includes information about biological processes, molecular functions and cellular components. The Reactome pathway database [32] was also used to identify pathway enrichment due to its database of human pathways, reactions and processes allowing an orientation and model in biological pathways that include classic intermediary metabolism, signaling, innate and adapted immunity, transcriptional regulation, apoptosis and diseases that were highly expressed in our data. Exclusive pathways were identified using InteractiVenn [94].

To visualize the list of GO terms and find how genes were functionally grouped, we used Cytoscape v.3.8.2 with the plugin ClueGO v.2.5.7 [37] with a (*p* < 0.001) and kappa statistics to calculate the relationships among the terms based on the similarity of their associated genes.

## 5. Conclusions

In summary, we used dimensionality reduction and a pathway analysis to integrate a large quantity of data in order to identify common genetic T-cell signatures across different type of cancers. These methodologies allowed us to compare those T-cell signatures and their core dynamics pathways between malignant and nonmalignant samples to identify unique and common pathways in CD4-T, CD8-T and Treg cells. Our analysis revealed that pathways related to the immune response, metabolism and viral immunoregulation were observed exclusively in cancer samples. Several other pathways were identified in all three T-cell subsets; however, future research is required to understand whether these pathways favor effective antitumor responses, or whether they are impaired and therefore do not prevent tumor progression.

## Figures and Tables

**Figure 1 ijms-23-11065-f001:**
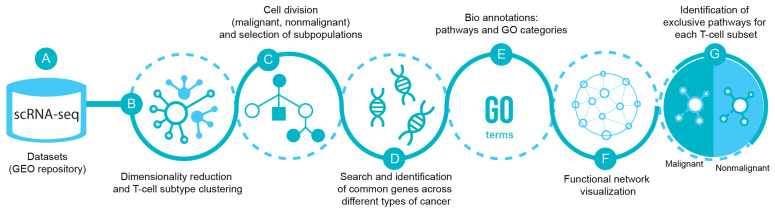
scRNA-seq pipeline. (**A**) Datasets of scRNA-seq were selected from Gene Expression Omnibus (GEO) database from breast (GSE114727 and GSE75688), lung (GSE126030 and GSE99254), colorectal (GSE108989), melanoma (GSE72056 and GSE123139), and head and neck cancer (GSE103322). (**B**) The data were processed using dimensionality reduction and clustering techniques in order to separate T-cells and their subpopulations CD4-T, CD8-T and Treg cells (**C**) from malignant and nonmalignant origin. (**D**) Then, common genes across the different types of cancer from malignant and nonmalignant origin were selected and (**E**) pathways and GO categories to profile the gene selection were obtained from Gene Ontology (GO) and Reactome databases. (**F**) Visualization using Cytoscape and ClueGO plugin for biological process and (**G**) differentiation of the functions among conditions of the subpopulations of T-cells were displayed.

**Figure 2 ijms-23-11065-f002:**
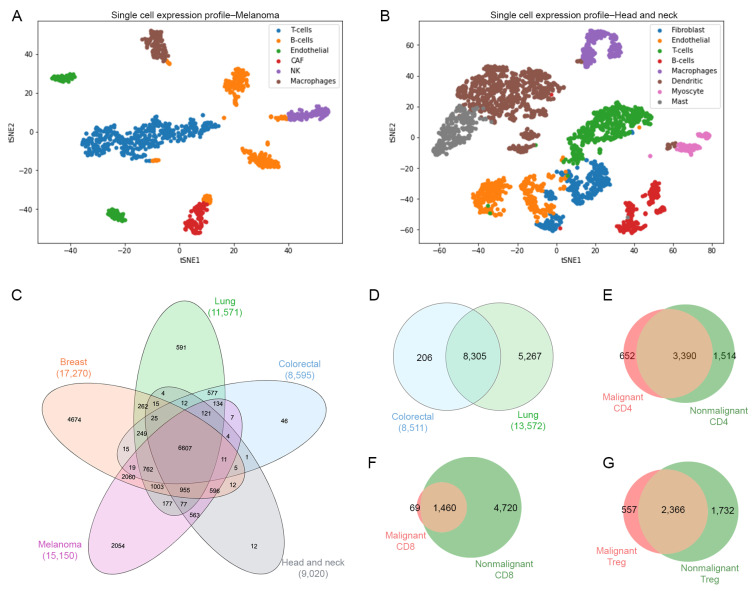
Cell identification. (**A**) Single cells plot of gene expression profiles using dimensionality reduction technique t-SNE for malignant cells for melanoma cancer data colored by type of cells such as T, B, macrophages, endothelial, CAF (cancer associated fibroblasts) and natural killer cells. (**B**) Malignant cells for head and neck cancer data colored by type of cells such as T, B/plasma, macrophages, dendritic, mast, endothelial, fibroblast, myocytes cells. (**C**) Venn diagram of number of genes of T-cells across different types of cancer. (**D**) Venn diagram of number of genes of T-cells from nonmalignant samples. Venn diagrams for (**E**) CD4-T cells, (**F**) CD8-T cells, and (**G**) Treg cells representing the exclusive and common genes between malignant and nonmalignant origin.

**Figure 3 ijms-23-11065-f003:**
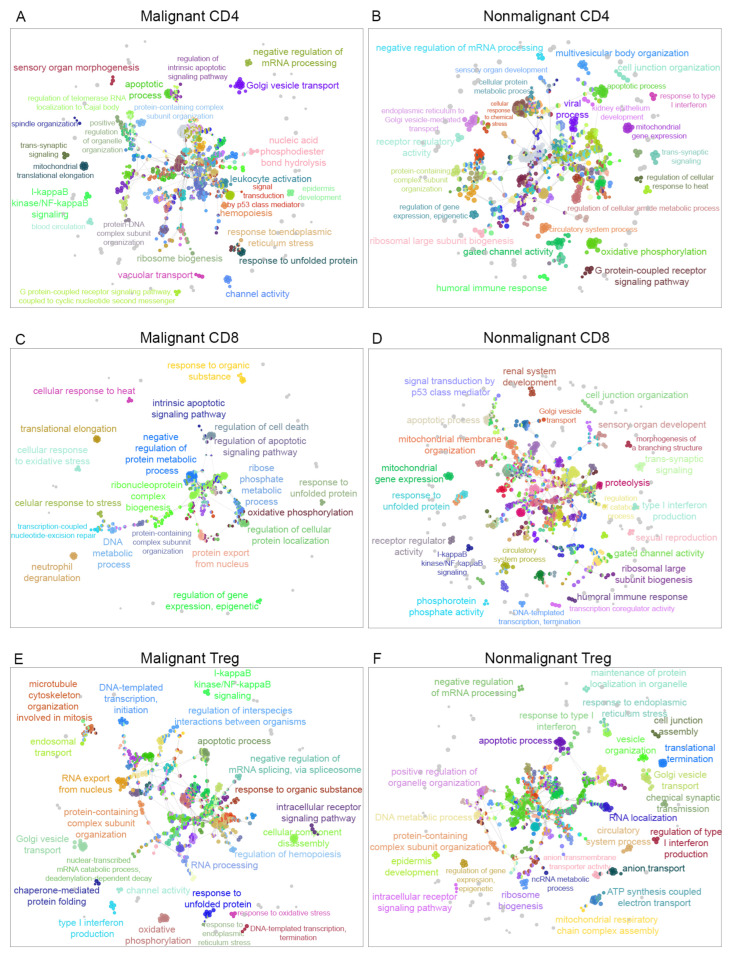
Gene ontology of biological processes networks. The visualization of the networks was performed in Cytoscape using the ClueGO plugin. (**A**) Network visualization of GO terms associated for CD4-T cells with malignant origin and (**B**) for GO terms associated with CD4-T cells from nonmalignant origin. (**C**) Network corresponding to GO terms of CD8-T cells from malignant origin and (**D**) network representing the GO terms associated with CD8-T cells with nonmalignant origin. (**E**) GO terms associated with Treg cells from malignant origin and (**F**) GO terms associated with Treg cells from nonmalignant origin.

**Figure 4 ijms-23-11065-f004:**
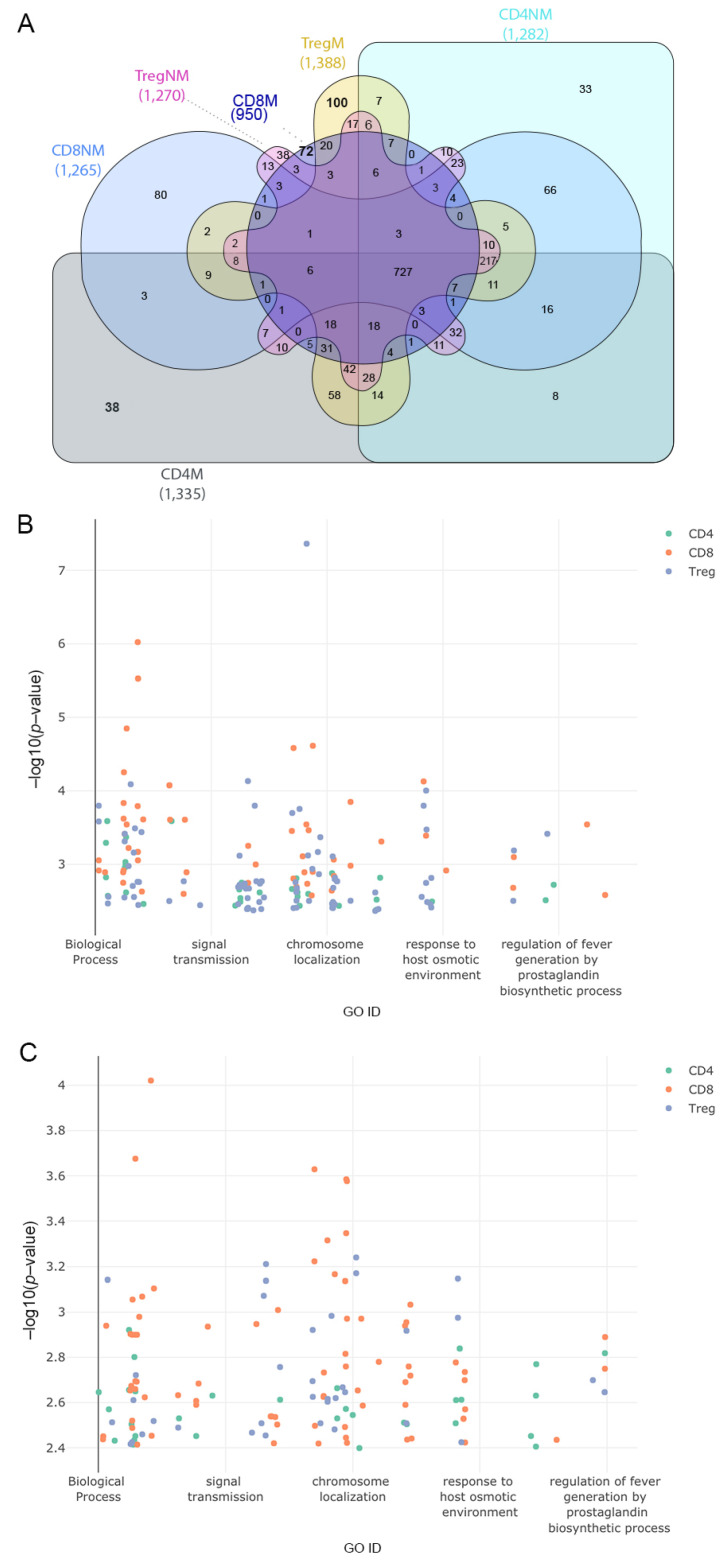
Gene ontology terms associated with exclusive pathways among all the samples. (**A**) Comparison of common biological processes across all the T-cell subpopulation. (**B**) Scatter plot of exclusive GO terms in biological processes in a two-dimensional space representing on the *x*-axis a semantic cross-match to the gene ontology database and on the *y*-axis the *p*-value (−log10 *p*-value) of the associated genes to the functions. Colors represent the subpopulation of the cell that correspond to the GO terms for samples with malignant origin. (**C**) Same as in (**B**) but for samples with nonmalignant origin. Interactive plots are available at https://tumourtcells.github.io/Fig-4B/ and https://tumourtcells.github.io/Fig-4C/ (accessed on 20 March 2021).

**Figure 5 ijms-23-11065-f005:**
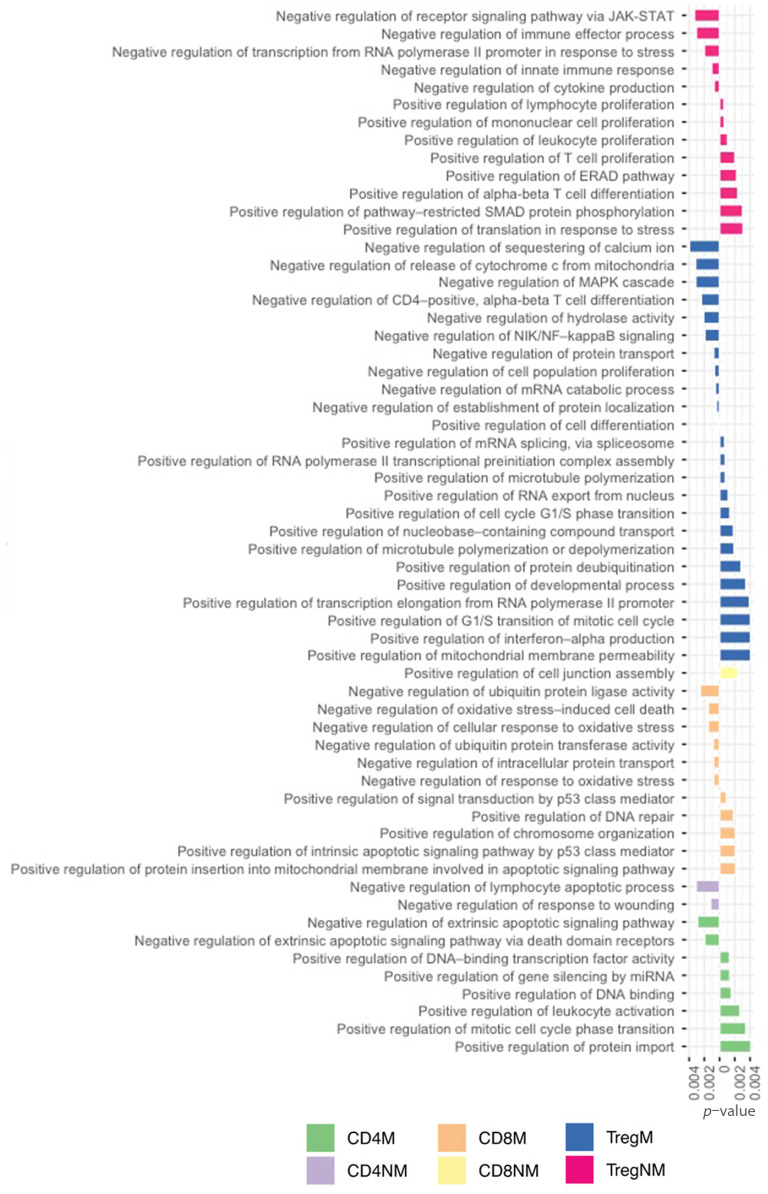
Exclusive GO terms for biological processes associated with positive and negative regulations to each T-cell subpopulation.

**Figure 6 ijms-23-11065-f006:**
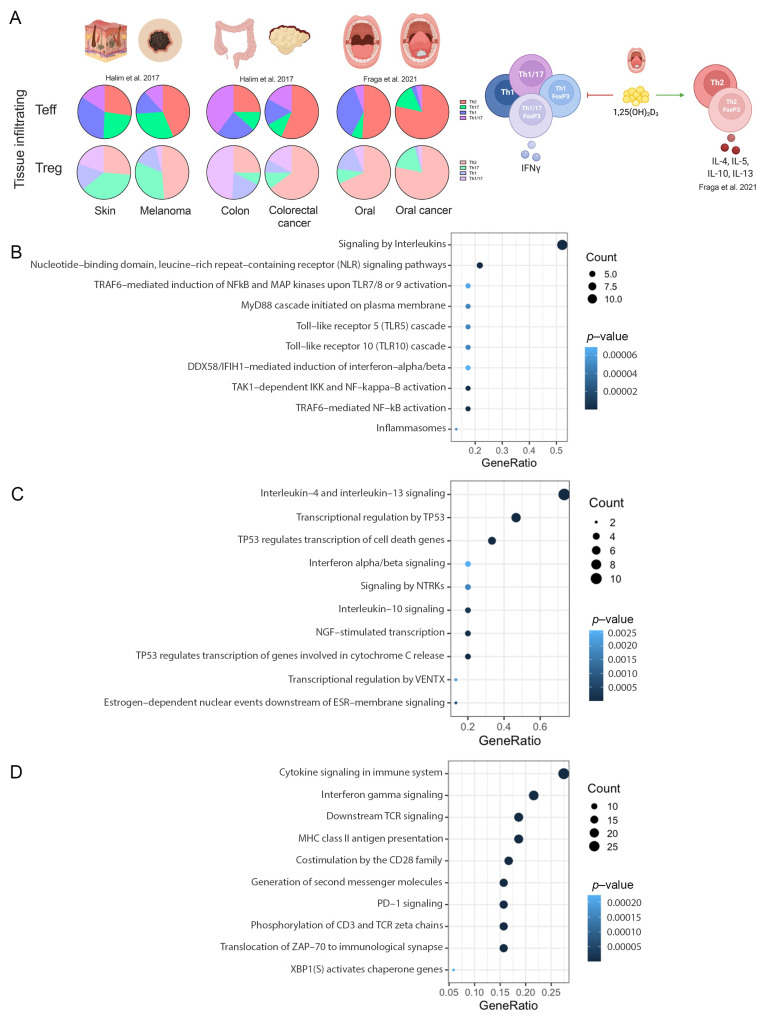
Pathways validation. (**A**) Summary of T-cells in melanoma, colorectal cancer and oral cancer from our lab [6,41]. Scatter plots of: (**B**) common pathways between CD4-T cells and RNA-seq experiments; (**C**) common pathways between Treg and RNA-seq experiments. (**D**) common pathways between CD4-T cells and proteomic data.

**Table 1 ijms-23-11065-t001:** Summary of the datasets used in this work. Each row indicates the number of genes for CD4-T, CD8-T and Treg cells.

Data ID	Type of Cancer	Condition	CD4	CD8	Treg
GSE114727	Breast	Malignant	12,855	12,601	8924
GSE75688	Breast	Malignant	8878	2620	6835
GSE126030	Lung	Nonmalignant	8418	10,573	6620
GSE99254	Lung	Malignant	10,571	11,034	10,349
		Nonmalignant	9748	10,488	7532
GSE108989	Colorectal	Malignant	7324	7005	7283
		Nonmalignant	6270	7280	6725
GSE103322	Head and neck	Malignant	7574	8093	7517
GSE72056	Melanoma	Malignant	10,136	10,779	8732
GSE123139	Melanoma	Malignant	10,579	10,906	7187

**Table 2 ijms-23-11065-t002:** Summary of functional enrichment annotations for malignant and nonmalignant CD4-T, CD8-T and Treg cells.

Functional Enrichment	Malignant			Nonmalignant		
	CD4	CD8	Treg	CD4	CD8	Treg
Biological process	1335	950	1388	1282	1265	1270
Molecular function	270	159	249	240	249	237
Cellular component	376	271	385	322	339	342
Reactome pathway	2059	1746	1966	2073	2138	2051

## Data Availability

Single cell RNA-seq samples included in our study were downloaded from the Gene Expression Omnibus (GEO) repository (https://www.ncbi.nlm.nih.gov/geo/, accessed on 15 February 2021) with the accession numbers: GSE114727, GSE75688, GSE126030, GSE99254, GSE108989, GSE103322, GSE72056 and GSE123139.

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
