# Peer review of "Analysis of Tumor-Infiltrating T-Cell Transcriptomes Reveal a Unique Genetic Signature across Different Types of Cancer"

_ijms, 2022, doi:10.3390/ijms231911065_

Round 1

Reviewer 1 Report

General comments:

Authors analyzed the single cell transcriptome of 5 tumor-infiltrating CD4-T, CD8-T and Tregs obtained from different types of cancer to identify specific pathways for each subset in malignant environments. The authors observed that cytokine signaling, especially Th2-type cytokine was the top overrepresented pathway in Tregs from malignant samples.

Authors undertook an extensive effort to collect and analyze data, which is very valuable. However, this manuscript is only descriptive and difficult to follow, which readers might consider not helpful.

Major comments:

1) Like some other bioinformatics papers, the messages in this manuscript are too scattered to digest despite the enormous amount of data. And each point is often not extensively explored. For example, a conclusion of 2.4 “In summary, our data revealed that despite the type of cancer, there are pathways in T-cells that are common between cancer, but unique in comparison with pathways in T-cells from non-malignant tissues.”(P12, line 283-285) would not help any further study. What are the novel biological observations which the authors wanted to deliver to readers throughout this manuscript?

2) In Introduction (P2 Line 37-43), the following sentence “Therefore, it is crucial to identify whether Tregs, CD4-T and CD8-T cells surrounding tumors exhibit a common specific genetic signature in comparison with tissue-resident T-cell subsets from healthy volunteers across different types of cancer.” is unclear to me. Why is it crucial to examine a common genetic signature with healthy samples? Exclusive signature in malignancy would be important. 

In addition,  the following sentence “This information will lead us to the mechanism by which the tumors command Tregs, CD4-T and CD8-T signaling pathways as well as the identification of potential specific responses aimed at predicting the efficacy of clinical therapies for cancer treatment” does not make sense. Why would a common genetic signature be a predictive biomarker of ICI response? Since those sentences which are supposed to deliver rationale for the current manuscript’s study are not connected logically at all in the Introduction part, the objectives of this manuscript are difficult to follow.

3) In Introduction (P2 Line 59-70), this paragraph may be unnecessary in Introduction. 

Minor comments:

1) Fig1 is very clear to understand.

Reviewer 2 Report

In the present work, the authors analyzed different tumor types to assess the differences between T cells such as CD4, CD8 and NK infiltrating the tumor and the healthy counterpart.  Therefore, the transcriptome of individual tumor-infiltrating CD4-T, CD8-T and Tregs cells obtained from different tumor types was analyzed to identify specific pathways for each subgroup. These were compared with nonmalignant genes for each subgroup of T cells to identify common pathways. Finally, the authors observed that cytokine signaling, particularly those specific to the TH2 response are the most represented in Tregs from tumor samples.

The work is interesting and well presented. 

The authors analyzed CD4, CD8 and Treg T cells, but different subsets of CD8 cells play important roles in cancer. The authors also have data on T cell subsets central and effector memory? Pathways and cytokines related to T cell activation (IFNg, GZMB, Perforin) have been analyzed? 

Round 2

Reviewer 1 Report

Authors have improved the quality of the manuscript. Although the content of the paper remains descriptive, I hope researchers would find some keys to extend their research from this paper.

This manuscript is a resubmission of an earlier submission. The following is a list of the peer review reports and author responses from that submission.